# Experience Replay with Likelihood-free Importance Weights

## Abstract

The use of past experiences to accelerate temporal difference (TD) learning of value functions, or experience replay, is a key component in deep reinforcement learning. In this work, we propose to reweight experiences based on their likelihood under the stationary distribution of the current policy, and justify this with a contraction argument over the Bellman evaluation operator. The resulting TD objective encourages small approximation errors on the value function over frequently encountered states. To balance bias and variance in practice, we use a likelihood-free density ratio estimator between on-policy and off-policy experiences, and use the ratios as the prioritization weights. We apply the proposed approach empirically on three competitive methods, Soft Actor Critic (SAC), Twin Delayed Deep Deterministic policy gradient (TD3) and Data-regularized Q (DrQ), over 11 tasks from OpenAI gym and DeepMind control suite. We achieve superior sample complexity on 35 out of 45 method-task combinations compared to the best baseline and similar sample complexity on the remaining 10.

## 1 Introduction

Deep reinforcement learning methods have achieved much success in a wide variety of domains (Mnih et al., 2016; Lillicrap et al., 2015; Horgan et al., 2018). While on-policy methods (Schulman et al., 2017) are effective, using off-policy data often yields better sample efficiency (Haarnoja et al., 2018; Fujimoto et al., 2018), which is critical when querying the environment is expensive and experiences are difficult to obtain. Experience replay (Lin, 1992) is a popular paradigm in off-policy reinforcement learning, where experiences stored in a replay memory can be reused to perform additional updates. When applied to temporal difference (TD) learning of the $Q$-value function (Mnih et al., 2015), the use of replay buffers avoids catastrophic forgetting of previous experiences and improves learning. Selecting experiences from the replay buffers using a prioritization strategy (instead of uniformly) can lead to large empirical improvements in terms of sample efficiency (Hessel et al., 2017).

Existing prioritization procedures rely on certain choices of importance sampling; for instance, Prioritized Experience Replay (PER) selects experiences with high TD error more often, and then down-weight the experiences that are frequently sampled in order to become closer to uniform sampling over the experiences (Schaul et al., 2015). However, this might not work well in actor-critic methods, where the goal is to learn the value function (or $Q$-value function) induced by the current policy, and following off-policy experiences might be harmful. In this case, it might be more beneficial to perform importance sampling that reflects on-policy experiences instead.

Based on this intuition, we investigate a new prioritization strategy for actor-critic methods based on the likelihood (i.e., the frequency) of experiences under the stationary distribution of the current policy (Tsitsiklis et al., 1997). In actor-critic methods (Konda & Tsitsiklis, 2000), we can estimate the value function of a policy by minimizing the expected squared difference between the critic network and its target value over a replay buffer; an appropriate replay buffer should properly reflect the discrepancy between critic value functions. We treat a discrepancy as "proper" if it preserves the contraction properties of the Bellman operator, and consider discrepancies measured by the expected squared distances under some state-action distribution. In Theorem 1 we prove that the stationary distribution of the current policy is the *only* distribution in which the Bellman operator is a contraction (i.e. being "proper"); this motivates the use of the stationary distribution as the underlying distribution for the replay buffer. Intuitively, optimizing the expected TD-error under the stationary distribution

addresses the TD-learning issue in actor-critic methods, as the TD errors in high-frequency states are given more weight.

To use replay buffers derived from the stationary distribution with existing deep reinforcement learning methods, we need to be mindful of the following bias-variance trade-off. We have fewer experiences from the current policy (using which results in high variance), but more experiences from other policies under the same environment (using which results in high bias). We propose to find appropriate bias-variance trade-offs by using importance sampling over the replay buffer, which requires an estimate of the density ratio between the stationary policy distribution and the replay buffer. Inspired by recent advances in inverse reinforcement learning (Fu et al., 2017) and off-policy policy evaluation (Grover et al., 2019), we use a likelihood-free method to obtain an estimate of the density ratio from a classifier trained to distinguish different types of experiences. We consider a smaller, "fast" replay buffer that contains near on-policy experiences, and a larger, "slow" replay buffer that contains additional off-policy experiences, and estimate density ratios between the two buffers. We then use these estimated density ratios as importance weights over the $Q$-value function update objective. This encourages more updates over state-action pairs that are more likely under the stationary policy distribution of the current policy, i.e., closer to the fast replay buffer.

Our approach can be readily combined with existing approaches that learn value functions from replay buffers. We consider our approach over three competitive actor-critic methods, Soft Actor-Critic (SAC, Haarnoja et al. (2018)), Twin Delayed Deep Deterministic policy gradient (TD3, Fujimoto et al. (2018)) and Data-regularized Q (DrQ, Kostrikov et al. (2020)). We demonstrate the effectiveness of our approach over on 11 environments from OpenAI gym (Dhariwal et al., 2017) and DeepMind Control Suite (Tassa et al., 2018), where both low-dimensional state space and high-dimensional image space are considered; this results in 45 method-task combinations in total. Notably, our approach outperforms the respective baselines in 35 out of the 45 cases, while being competitive in the remaining 10 cases. This demonstrates that our method can be applied as a simple plug-and-play approach to improve existing actor-critic methods.

## 2    PRELIMINARIES

The reinforcement learning problem can be described as finding a policy for a Markov decision process (MDP) defined as the following tuple $(\mathcal{S}, \mathcal{A}, P, r, \gamma, p_0)$, where $\mathcal{S}$ is the state space, $\mathcal{A}$ is the action space, $P : \mathcal{S} \times \mathcal{A} \to \mathcal{P}(\mathcal{S})$ is the transition kernel, $r : \mathcal{S} \times \mathcal{A} \to \mathbb{R}$ is the reward function, $\gamma \in [0, 1)$ is the discount factor and $p_0 \in \mathcal{P}(\mathcal{S})$ is the initial state distribution. The goal is to learn a stationary policy $\pi : \mathcal{S} \to \mathcal{P}(\mathcal{A})$ that selects actions in $\mathcal{A}$ for each state $s \in \mathcal{S}$, such that the policy maximizes the expected sum of rewards: $J(\pi) := \mathbb{E}_\pi \left[ \sum_{t=0}^\infty \gamma^t r(s_t, a_t) \right]$, where the expectation is over trajectories sampled from $s_0 \sim p_0$, $a_t \sim \pi(\cdot|s_t)$, and $s_{t+1} \sim P(\cdot|s_t, a_t)$ for $t \geq 0$.

For a fixed policy, the MDP becomes a Markov chain, so we define the state-action distribution at timestep $t$: $d_t^\pi(s, a)$, and the the corresponding (unnormalized) stationary distribution over states and actions $d_\pi(s, a) = \sum_{t=0}^\infty \gamma^t d_t^\pi(s, a)$ (we assume this always exists for the policies we consider). We can then write $J(\pi) = \mathbb{E}_{d^\pi}[r(s, a)]$. For any stationary policy $\pi$, we define its corresponding state-action value function as $Q^\pi(s, a) := \mathbb{E}_\pi[\sum_{t=0}^\infty \gamma^t r(s_t, a_t)|s_0 = s, a_0 = a]$, its corresponding value function as $V^\pi(s) := \mathbb{E}_{a \sim \pi(\cdot|s)}[Q^\pi(s, a)]$ and the advantage function $A^\pi(s, a) = Q^\pi(s, a) - V^\pi(s)$. A large variety of actor-critic methods (Konda & Tsitsiklis, 2000) have been developed in the context of deep reinforcement learning (Silver et al., 2014; Mnih et al., 2016; Lillicrap et al., 2015; Haarnoja et al., 2018; Fujimoto et al., 2018), where learning good approximations to the $Q$-function is critical to the success of any deep reinforcement learning method based on actor-critic paradigms.

The $Q$-function can be learned via temporal difference (TD) learning (Sutton, 1988) based on the Bellman equation $Q^\pi(s, a) = \mathcal{B}^\pi Q^\pi(s, a)$; where $\mathcal{B}^\pi$ denotes the Bellman evaluation operator

$$\mathcal{B}^\pi Q(s, a) := r(s, a) + \gamma \mathbb{E}_{s', a'}[Q(s', a')], \qquad (1)$$

where in the expectation we sample the next step, $s' \sim P(\cdot|s, a)$ and $a' \sim \pi(\cdot|s)$. Given some experience replay buffer $\mathcal{D}$ (collected by navigating the same environment, but with unknown and potentially different policies), one could optimize the following loss for a $Q$-network:

$$L_Q(\theta; \mathcal{D}) = \mathbb{E}_{(s,a) \sim \mathcal{D}} \left[ (Q_\theta(s, a) - \hat{\mathcal{B}}^\pi Q_\theta(s, a))^2 \right] \qquad (2)$$

which fits $Q_\theta(s, a)$ to an estimate of the target value $\hat{\mathcal{B}}^\pi[Q_\theta](s, a)$[1]. In practice, the target values can be estimated either via on-policy experiences (Sutton et al., 1999) or via off-policy experiences (Precup, 2000; Munos et al., 2016).

Ideally, we can learn $Q^\pi$ by optimizing the $L_Q(\theta; \mathcal{D})$ to zero with over-parametrized neural networks. However, instead of minimizing the loss $L_Q(\theta; \mathcal{D})$ directly, prioritization over the sampled replay buffer $\mathcal{D}$ could lead to stronger performance. For example, prioritized experience replay (PER, (Schaul et al., 2015)) is a heuristic that assigns higher weights to transitions with higher TD errors, and is applied successfully in deep $Q$-learning (Hessel et al., 2017).

## 3   PRIORITIZED EXPERIENCE REPLAY BASED ON STATIONARY DISTRIBUTIONS

Assume that $d$, the distribution the replay buffer $\mathcal{D}$ is sampled from, is supported on the entire space $\mathcal{S} \times \mathcal{A}$, and that we have infinite samples from $\pi$ (so the Bellman target is unbiased). Let us define the TD-learning objective for $Q$ with prioritization weights $w : \mathcal{S} \times \mathcal{A} \to \mathbb{R}^+$, under the sampling distribution $d \in \mathcal{P}(\mathcal{S} \times \mathcal{A})$:

$$L_Q(\theta; d, w) = \mathbb{E}_d \left[ w(s, a)(Q_\theta(s, a) - \mathcal{B}^\pi Q_\theta(s, a))^2 \right] \tag{3}$$

In practice, the expectation in $L_Q(\theta; d, w)$ can be estimated with Monte-Carlo methods, such as importance sampling, rejection sampling, or combinations of multiple methods (such as in PER (Schaul et al., 2015)). Without loss of generality, we can treat the problem as optimizing the mean squared TD error under some *priority distribution* $d^w \propto d \cdot w$, since:

$$\arg\min_\theta L_Q(\theta; d, w) = \arg\min_\theta L_Q(\theta; d^w), \tag{4}$$

so one could treat prioritized experience replay for TD learning as selecting a favorable *priority distribution* $d^w$ (under which the $L_Q$ loss is computed) in order to improve some notion of performance.

In this paper, we propose to use as *priority distribution* $d^w = d^\pi$, where $d^\pi$ is the stationary distribution of state-action pairs under the current policy $\pi$. This reflects the intuition that TD-errors in high-frequency state-action pairs are more problematic than in low-frequency ones, as they will negatively impact policy updates more severely. In the following subsection, we argue the importance of choosing $d^\pi$ from the perspective of maintaining desirable contraction properties of the Bellman operators under more general norms. If we consider Euclidean norms weighted under some distribution $d^w \in \mathcal{P}(\mathcal{S} \times \mathcal{A})$, the usual $\gamma$-contraction argument for Bellman operators holds only for $d^w = d^\pi$, and not for other distributions.

### 3.1   POLICY-DEPENDENT NORMS FOR BELLMAN BACKUP

The convergence of Bellman updates relies on the fact that the Bellman evaluation operator $\mathcal{B}^\pi$ is a $\gamma$-contraction with respect to the $\ell_\infty$ norm, i.e. $\forall Q, Q' \in \mathcal{Q}$, where $\mathcal{Q} = \{Q : (\mathcal{S} \times \mathcal{A}) \to \mathbb{R}\}$ is the set of all possible $Q$ functions:

$$\|\mathcal{B}^\pi Q - \mathcal{B}^\pi Q'\|_\infty \leq \gamma \|Q - Q'\|_\infty \tag{5}$$

While it is sufficient to show convergence results, the $\ell_\infty$ norm reflects a distance over two $Q$ functions under the worst possible state-action pair, and is independent of the current policy. If two $Q$ functions are equal everywhere except for a large difference on a single state-action pair $(\tilde{s}, \tilde{a})$ that is unlikely under $d^\pi$, the $\ell_\infty$ distance between the two $Q$ functions is large. In practice, however, this will have little effect over policy updates as it is unlikely for the current policy to sample $(\tilde{s}, \tilde{a})$.

Since our goal with the TD updates is to learn $Q^\pi$, a distance metric that is related to $\pi$ is a more suitable one for comparing different $Q$ functions, reflecting the intuition that errors in frequent state-action pairs are more costly than on infrequent ones. Let us consider the following weighted $\ell_2$ distance between $Q$ functions,

$$\|Q - Q'\|_d^2 := \mathbb{E}_{(s,a)\sim d}[(Q(s, a) - Q'(s, a))^2] \tag{6}$$

---

[1]We also do not take the gradient over the target, which is the more conventional approach.

where $d \in \mathcal{P}(\mathcal{S} \times \mathcal{A})$ is a distribution over state-action pairs. This can be treated as the $\ell_2$ norm but measured over stationary distribution $d$ as opposed to the Lebesgue measure. This is closely tied to the $L_Q$ objective since

$$L_Q(\theta; d) = \|Q_\theta(s, a) - \mathcal{B}^\pi Q_\theta(s, a)\|_d^2$$

In the following statements, we show that $\mathcal{B}^\pi$ is only a contraction operator when under the $\|\cdot\|_{d^\pi}$ norm; this supports the use of $d^\pi$ instead of other distributions for the $L_Q$ objective, as it reflects a more reasonable measurement of distance between $Q$-functions for policy $\pi$.

**Lemma 1.** *For all $\gamma \in (0, 1)$, the Bellman operator $\mathcal{B}^\pi$ is a $\gamma$-contraction with respect to the $\|\cdot\|_d$ norm if $d = d^\pi$ holds almost everywhere, i.e.,*

$$d = d^\pi \ a.e. \implies \|\mathcal{B}^\pi Q - \mathcal{B}^\pi Q'\|_d \leq \gamma \|Q - Q'\|_d, \forall Q, Q' \in \mathcal{Q}$$

*Proof.* In Appendix B. On a high-level, we apply Jensen's inequality to $f(x) = x^2$. □

**Theorem 1.** *For all $\gamma \in (0, 1)$, the Bellman operator $\mathcal{B}^\pi$ is a $\gamma$-contraction with respect to the $\|\cdot\|_d$ norm if and only if $d = d^\pi$ holds almost everywhere, i.e.,*

$$d = d^\pi, \ a.e. \iff \|\mathcal{B}^\pi Q - \mathcal{B}^\pi Q'\|_d \leq \gamma \|Q - Q'\|_d, \forall Q, Q' \in \mathcal{Q}$$

*Proof.* In Appendix B. On a high-level, whenever $d = d^\pi$ does not hold over some non-empty open set, we can perturb a constant $Q$-value function over this set to contradict $\gamma$-contraction. □

Theorem 1 highlights the importance of using $d^\pi$ in the $\|\cdot\|_d$ norm specifically for measuring the distance between $Q$-functions: if we use any distribution other than $d^\pi$, the Bellman operator is not guaranteed to be a $\gamma$-contraction under that distance, which leads to worse convergence rates.

## 3.2 TD Learning based on $d^\pi$

The $\|\cdot\|_{d^\pi}$ norm also captures our intuition that errors in high-frequency state-action pairs are more problematic than low-frequency ones, as they are likely to have larger effect in policy learning. For example, for the actor-critic policy gradient with $Q_\theta$:

$$\nabla_\phi J(\pi_\phi) = \mathbb{E}_{d^\pi}[\nabla_\phi \log \pi_\phi(a|s) Q_\theta(s, a)]; \tag{7}$$

if $(Q_\theta(s, a) - Q^\pi(s, a))^2$ is large for high-frequency $(s, a)$ tuples, then the policy update is likely to be worse than the update with ground truth $Q^\pi$. Moreover, the gradient descent update over the objective $L_Q(\theta; d^\pi)$:

$$\theta \leftarrow \theta - \eta \nabla_\theta L_Q(\theta; d^\pi) = \theta - \eta \mathbb{E}_{d^\pi}[(Q_\theta(s, a) - \hat{\mathcal{B}}^\pi Q_\theta(s, a)) \nabla_\theta Q_\theta(s, a)]$$

corresponds to a batch version of TD update. This places more emphasis on TD errors for state-action pairs that occur more frequently under the current policy.

To illustrate the validity of using $d^\pi$, we consider a chain MDP example (Figure 1a, but with 5 states in total), where the agent takes two actions that progress to the state on the left or on the right. The agent receives a final reward of 1 at the right-most state and rewards of 0 at other states. The policy takes the right action at each state with probability $p$, and takes the left action for with probability $1 - p$. We initialize the $Q$-function from $[0, 1]$ uniformly at random and consider $p = 0.8$ and $0.2$.

We compare three approaches to prioritization with TD updates: uniform over all state-action pairs, prioritization over TD error (as done in Schaul et al. (2015)), and prioritization with $d^\pi$; we include more details in Appendix C. We illustrate the $\|\cdot\|_{d^\pi}^2$ distance between the learned $Q$-function and the ground truth $Q$-function in Figure 1b; prioritization with $d^\pi$ outperforms both uniform sampling and prioritization with TD error in terms of speed of converging to the ground truth, especially at the initial iterations. When $p = 0.8$, $d^\pi$ only takes 120 steps on average to decrease the expected error to be smaller than 1, while TD error takes 182 steps on average; this means that prioritization with $d^\pi$ is helpful when we have a limited update budget.

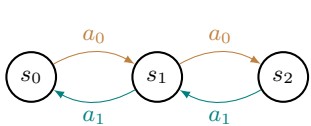

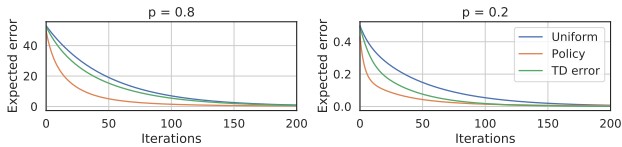

(a) Illustration of the chain MDP.  (b) Estimation error ($\mathbb{E}_{d^\pi}[(Q_\theta - Q^\pi)^2]$) for different prioritization methods, including uniform sampling (Uniform), sampling based on TD error (TD error), and sampling based on $d^\pi$ (Policy).

Figure 1: Simulation of TD updates with different prioritization methods.

## 4   LIKELIHOOD-FREE IMPORTANCE WEIGHTING OVER REPLAY BUFFERS

In practice, however, there are two challenges with regards to using $L_Q(\theta; d^\pi)$ as the objective. On the one hand, an accurate estimate of $d^\pi$ requires many on-policy samples from $d^\pi$ and interactions with the environment, which could increase the practical sample complexity; on the other hand, if we instead use off-policy experiences from the replay buffer, it would be difficult to estimate the importance ratio $w(s, a) := d^\pi(s, a)/d^D(s, a)$ when the replay buffer $\mathcal{D}$ is a mixture of trajectories from different policies. Therefore, likelihood-free density ratio estimation methods that rely only on samples (e.g. from the replay buffer) rather than likelihoods are more general and well-suited for estimating the objective function $L_Q(\theta; d^\pi)$ with a good bias-variance trade-off. An appropriate choice of importance weights allows us to balance bias (which comes from replay experiences of other policies) and variance (which comes from a small number of on-policy experiences).

In this paper, we use the variational representation of $f$-divergences (Csiszar, 1964) to estimate the density ratios. For any convex, lower-semicontinuous function $f : [0, \infty) \to \mathbb{R}$ satisfying $f(1) = 0$, the $f$-divergence between two probabilistic measures $P, Q \in \mathcal{P}(\mathcal{X})$ (where we assume $P \ll Q$, i.e. $P$ is absolutely continuous w.r.t. $Q$) is defined as: $D_f(P\|Q) = \int_\mathcal{X} f(\mathrm{d}P(\boldsymbol{x})/\mathrm{d}Q(\boldsymbol{x}))\,\mathrm{d}Q(\boldsymbol{x})$. A general variational method can be used to estimate $f$-divergences given only samples from $P$ and $Q$.

**Lemma 2** (Nguyen et al. (2008)). *Assume that $f$ has first order derivatives $f'$ at $[0, +\infty)$. $\forall P, Q \in \mathcal{P}(\mathcal{X})$ such that $P \ll Q$ and $w : \mathcal{X} \to \mathbb{R}^+$,*

$$D_f(P\|Q) \geq \mathbb{E}_P[f'(w(\boldsymbol{x}))] - \mathbb{E}_Q[f^*(f'(w(\boldsymbol{x})))] \tag{8}$$

*where $f^*$ denotes the convex conjugate and the equality is achieved when $w = \mathrm{d}P/\mathrm{d}Q$.*

We can apply this approach to estimating the density ratio $w(s, a) := d^\pi(s, a)/d^D(s, a)$ with samples from the replay buffer. These ratios are then multiplied to the $Q$-function updates to perform importance weighting. Specifically, we consider sampling from two types of replay buffers. One is the *regular (slow) replay buffer*, which contains a mixture of trajectories from different policies; the other is a *smaller (fast) replay buffer*, which contains only a small set of trajectories from very recent policies. After each episode of environment interaction, we update both replay buffers with the new experiences; the distribution of the slow replay buffer changes more slowly due to the larger size (hence the name "slow").

The slow replay buffer contains off-policy samples from $d^D$ whereas the fast replay buffer contains (approximately) on-policy samples from $d^\pi$ (assuming the buffer size is small enough). Therefore, the slow replay buffer has better coverage of transition dynamics of the environment while being less on-policy. Denoting the fast and slow replay buffers as $\mathcal{D}_\mathrm{f}$ and $\mathcal{D}_\mathrm{s}$ respectively, we estimate the ratio $d^\pi/d^D$ via minimizing the following objective over the network $w_\psi(\boldsymbol{x})$ parametrized by $\psi$ (the outputs $w_\psi(s, a)$ are forced to be non-negative via activation functions):

$$L_w(\psi) := \mathbb{E}_{\mathcal{D}_\mathrm{s}}[f^*(f'(w_\psi(s, a)))] - \mathbb{E}_{\mathcal{D}_\mathrm{f}}[f'(w_\psi(s, a))] \tag{9}$$

From Lemma 2, we can recover an estimate of the density ratio from the optimal $w_\psi$ by minimizing the $L_w(\psi)$ objective. To address the finite sample size issue, we apply self-normalization (Cochran, 2007) to the importance weights over the slow replay buffer $\mathcal{D}_\mathrm{s}$ with a hyperparameter $T$:

$$\tilde{w}_\psi(s, a) := \frac{w_\psi(s, a)^{1/T}}{\mathbb{E}_{\mathcal{D}_\mathrm{s}}[w_\psi(s, a)^{1/T}]} \tag{10}$$

The final objective for TD learning over $Q$ is then

$$L_Q(\theta; d^\pi) \approx L_Q(\theta; \mathcal{D}_s, \tilde{w}_\psi) := \mathbb{E}_{(s,a)\sim\mathcal{D}_s}[\tilde{w}_\psi(\boldsymbol{x})(Q_\theta(s,a) - \hat{\mathcal{B}}^\pi Q_\theta(s,a))^2] \qquad (11)$$

where the target $\hat{\mathcal{B}}^\pi Q_\theta$ is estimated via Monte Carlo samples. We keep the remainder of the algorithm, such as policy gradient and value network update (if available) unchanged, so this method can be adapted for different off-policy actor-critic algorithms, utilizing their respective advantages. We observe that using the weights to correct the policy updates does not provide much marginal improvements, so we did not consider this for comparison. We describe a general procedure of our approach in Algorithm 1 (Appendix A), where one can modify from some "base" actor-critic algorithm to implement our approach. These algorithm cover both stochastic and deterministic policies, as our method does not require likelihood estimates from the policy. We consider our divergences to be Jensen-Shannon, so $w_\psi$ can be treated as a probabilistic classifier.

## 5    RELATED WORK

Experience replay (Lin, 1992) is a crucial component in deep reinforcement learning (Hessel et al., 2017; Andrychowicz et al., 2017; Schaul et al., 2015), where off-policy experiences are utilized to improve sample efficiency. These experiences can be utilized on policy updates (such as in actor-critic methods (Konda & Tsitsiklis, 2000; Wang et al., 2016)), on value updates (such as in deep Q-learning (Schaul et al., 2015)) or on evaluating TD update targets (Precup, 2000; Precup et al., 2001). For value updates, there are two sources of randomness that could benefit from importance weights (prioritization). The first source is the evaluation of the TD learning target for longer traces such as TD($\lambda$); importance weights can be used to debias targets computed from off-policy trajectories (Precup, 2000; Munos et al., 2016; Espeholt et al., 2018; Schmitt et al., 2019), similar to its role in policy learning. The second source is the sampling of state-action pairs where the values are updated (Schaul et al., 2015), which is addressed in this paper.

Numerous techniques have achieved superior sample complexity through prioritization of replay buffers. In model-based planning, Prioritized Sweeping (Moore & Atkeson, 1993; Andre et al., 1998; van Seijen & Sutton, 2013) selects the next state updates according to changes in value. Prioritized Experience Replay (PER, (Schaul et al., 2015)) emphasizes experiences with larger TD errors and is critical to the success of sample efficient deep Q-learning (Hessel et al., 2017). Remember and Forget Experience Replay (ReF-ER, (Novati & Koumoutsakos, 2018)) removes the experiences if it differs much from choices of the current policy; this encourages sampling on-policy behavior which is similar to what we propose. Differing from ReF-ER, we do not require knowledge of the policy distribution. Distribution Correction (DisCor, Kumar et al. (2020)) suggests against using on-policy experiences, which seems to be in contrast to what we have promoted. However, their analysis is based on the Bellman *optimality* operator, which aims to find the optimal $Q$-value function, while ours is based on the Bellman *evaluation* operator, which aims to find the $Q$-value function under the current policy; this could partially explain why DisCor did not achieve superior performance than the baseline approach on OpenAI gym tasks.

Likelihood-free density ratio estimation have been adopted in imitation learning Ho & Ermon (2016), inverse reinforcement learning (Fu et al., 2017) and model-based off-policy policy evaluation (Grover et al., 2019). Different from these cases, we do not use the weights to estimate the advantage function or to reduce bias in reward estimation; our goal is to improve performance of TD learning with function approximation. Dual representations of $f$-divergences are also leveraged in reinforcement learning (Nachum et al., 2019; Nachum & Dai, 2020), but it is used over a regularizer that encourages exploration to be closer to off-policy experiences; the importance weights are added to the reward function when computing the $Q$-value function but do not affect the replay experiences otherwise.

## 6    EXPERIMENTS

We combine the proposed prioritization approach over three popular actor-critic algorithms, namely Soft-Actor Critic (SAC, Haarnoja et al. (2018)), Twin Delayed Deep Deterministic policy gradient (TD3, Fujimoto et al. (2018)) and Data-regularized Q (DrQ, Kostrikov et al. (2020)). We compare our method with alternative approaches to prioritization; these include uniform sampling over the replay buffer (adopted by the original SAC and TD3 methods) and prioritization experience replay

Table 1: Results on OpenAI Gym environments when trained with 500k steps. ERE is only designed for SAC, so its results on TD3 are not available.

| SAC based | SAC | +PER | +ERE | +LFIW |
|---|---|---|---|---|
| Ant-v2 | $3193 \pm 404$ | $2764 \pm 287$ | $3331 \pm 298$ | $\mathbf{3579} \pm 260$ |
| HalfCheetah-v2 | $8325 \pm 408$ | $8111 \pm 341$ | $8631 \pm 189$ | $\mathbf{9045} \pm 222$ |
| Hopper-v2 | $2645 \pm 310$ | $2871 \pm 214$ | $2512 \pm 301$ | $\mathbf{3109} \pm 244$ |
| Humanoid-v2 | $2033 \pm 199$ | $1459 \pm 208$ | $2466 \pm 147$ | $\mathbf{3189} \pm 231$ |
| Walker2d-v2 | $2914 \pm 189$ | $3071 \pm 109$ | $2990 \pm 217$ | $\mathbf{3221} \pm 149$ |
| **TD3 based** | **TD3** | **+PER** | **+ERE** | **+LFIW** |
| Ant-v2 | $2663 \pm 372$ | $2610 \pm 128$ | | $\mathbf{2990} \pm 178$ |
| HalfCheetah-v2 | $7527 \pm 438$ | $7310 \pm 339$ | N/A | $\mathbf{8567} \pm 491$ |
| Hopper-v2 | $1801 \pm 206$ | $\mathbf{2019} \pm 109$ | | $1937 \pm 250$ |
| Walker2d-v2 | $1306 \pm 257$ | $1241 \pm 122$ | | $\mathbf{2113} \pm 310$ |

Table 2: Results of SAC and TD3 trained from states on the DeepMind Control environments with and without LFIW after 100k and 250k environment steps. **The results show significant improvements when the agents is trained with LFIW.** Results are reported over 5 random seeds. The maximum possible score for any environment is 1,000.

| Training steps | 100k | | | 250k | | |
|---|---|---|---|---|---|---|
| **TD3 based** | **TD3** | **+PER** | **+LFIW** | **TD3** | **+PER** | **+LFIW** |
| Finger, Spin | $315 \pm 49$ | $306 \pm 68$ | $\mathbf{467} \pm 49$ | $654 \pm 16$ | $587 \pm 32$ | $\mathbf{788} \pm 43$ |
| Cartpole, Swing | $464 \pm 50$ | $457 \pm 30$ | $\mathbf{567} \pm 28$ | $750 \pm 21$ | $702 \pm 21$ | $\mathbf{801} \pm 39$ |
| Reacher, Easy | $404 \pm 73$ | $351 \pm 109$ | $\mathbf{592} \pm 90$ | $711 \pm 21$ | $709 \pm 56$ | $\mathbf{892} \pm 45$ |
| Cheetah, Run | $\mathbf{108} \pm 20$ | $\mathbf{98} \pm 23$ | $\mathbf{113} \pm 18$ | $381 \pm 22$ | $390 \pm 46$ | $\mathbf{414} \pm 20$ |
| Walker, Walk | $349 \pm 22$ | $336 \pm 18$ | $\mathbf{451} \pm 45$ | $765 \pm 31$ | $724 \pm 40$ | $\mathbf{811} \pm 20$ |
| Ball in Cup, Catch | $278 \pm 29$ | $213 \pm 11$ | $\mathbf{418} \pm 44$ | $664 \pm 21$ | $687 \pm 24$ | $\mathbf{872} \pm 27$ |

Table 3: Results of SAC and TD3 trained from states on the DeepMind Control environments with and without LFIW after 100k and 250k environment steps. **The results show significant improvements when the agents is trained with LFIW.** Results are reported over 5 random seeds. The maximum possible score for any environment is 1,000.

| 100k environment steps | | | | |
|---|---|---|---|---|
| **SAC based** | **SAC** | **+PER** | **+PER+LFIW** | **+LFIW** |
| Finger, Spin | $482 \pm 34$ | $486 \pm 18$ | $503 \pm 27$ | $\mathbf{523} \pm 16$ |
| Cartpole, Swing | $700 \pm 51$ | $689 \pm 39$ | $726 \pm 14$ | $\mathbf{789} \pm 27$ |
| Reacher, Easy | $750 \pm 68$ | $704 \pm 89$ | $806 \pm 55$ | $\mathbf{861} \pm 29$ |
| Cheetah, Run | $498 \pm 108$ | $367 \pm 123$ | $502 \pm 109$ | $\mathbf{541} \pm 89$ |
| Walker, Walk | $187 \pm 89$ | $234 \pm 31$ | $\mathbf{321} \pm 29$ | $\mathbf{333} \pm 12$ |
| Ball in Cup, Catch | $\mathbf{888} \pm 13$ | $834 \pm 23$ | $\mathbf{892} \pm 8$ | $\mathbf{890} \pm 6$ |
| 250k environment steps | | | | |
| **SAC based** | **SAC** | **+PER** | **+PER+LFIW** | **+LFIW** |
| Finger, Spin | $806 \pm 47$ | $814 \pm 45$ | $860 \pm 23$ | $\mathbf{901} \pm 14$ |
| Cartpole, Swing | $825 \pm 8$ | $811 \pm 15$ | $823 \pm 31$ | $\mathbf{873} \pm 23$ |
| Reacher, Easy | $\mathbf{945} \pm 32$ | $931 \pm 11$ | $\mathbf{944} \pm 6$ | $\mathbf{941} \pm 21$ |
| Cheetah, Run | $638 \pm 32$ | $618 \pm 41$ | $631 \pm 56$ | $\mathbf{709} \pm 11$ |
| Walker, Walk | $895 \pm 47$ | $881 \pm 23$ | $\mathbf{917} \pm 20$ | $\mathbf{911} \pm 12$ |
| Ball in Cup, Catch | $\mathbf{974} \pm 13$ | $\mathbf{978} \pm 7$ | $\mathbf{970} \pm 7$ | $\mathbf{981} \pm 19$ |

based on TD-error (Schaul et al., 2015). We choose 5 continuous control tasks from the OpenAI gym

Table 4: Results for DrQ (Kostrikov et al., 2020) on the image-based RL on the DeepMind Control Suite. LFIW is applied to a state-of-the-art image-based RL algorithm in DrQ, and we are able to see consistent improvement over the DM Control Suite Benchmark.

| 100k steps | DrQ | DrQ+LFIW | 500k steps | DrQ | DrQ+LFIW |
|---|---|---|---|---|---|
| Finger, Spin | $838 \pm 58$ | $\mathbf{909} \pm 28$ | Finger, Spin | $\mathbf{918} \pm 49$ | $\mathbf{922} \pm 28$ |
| Cartpole, Swing | $748 \pm 50$ | $\mathbf{801} \pm 22$ | Cartpole, Swing | $875 \pm 6$ | $\mathbf{893} \pm 8$ |
| Reacher, Easy | $573 \pm 67$ | $\mathbf{743} \pm 89$ | Reacher, Easy | $\mathbf{945} \pm 25$ | $\mathbf{939} \pm 12$ |
| Cheetah, Run | $387 \pm 45$ | $\mathbf{444} \pm 38$ | Cheetah, Run | $574 \pm 104$ | $\mathbf{581} \pm 112$ |
| Walker, Walk | $639 \pm 99$ | $\mathbf{718} \pm 86$ | Walker, Walk | $901 \pm 35$ | $\mathbf{909} \pm 38$ |
| Ball in Cup, Catch | $\mathbf{901} \pm 17$ | $\mathbf{901} \pm 25$ | Ball in Cup, Catch | $\mathbf{970} \pm 4$ | $968 \pm 8$ |

benchmark (Brockman et al., 2016) and 6 tasks from the DeepMind Control suite (DCS, Tassa et al. (2018)). We consider state representations in all tasks and pixel representations from DCS.

Our method introduces some additional hyperparameters compared to the vanilla approaches, namely the temperature $T$, the size of the fast replay buffer $|\mathcal{D}_\mathrm{f}|$ and the architecture for the density estimator $w_\psi$. To ensure fair comparisons against the baselines, we use the same hyperparameters as the original algorithms when it is available. For all environments we use the following default hyperparameters for likelihood-free importance weighting: $T = 5$, $|\mathcal{D}_\mathrm{f}| = 10^4$, $|\mathcal{D}_\mathrm{s}| = 10^6$. We use $f$ from the Jensen Shannon divergence for better numerical stability. We include more experimental details in Appendix C.

## 6.1 EVALUATION

We use (+LFIW) to denote our likelihood-free importance weighting method, (+PER) to denote prioritization with TD error (Schaul et al., 2015)[2] and (+ERE) to denote Emphasizing Recent Experience (ERE, Wang & Ross (2019)) for SAC only. Table 1 shows the results on OpenAI gym (500k steps), whereas Tables 2 and 4 shows the results on DMCS with state (100k and 250k steps) and image representations (100k and 500k steps) respectively. These steps are chosen to demonstrate both initial training progress and approximate performance at convergence.

**OpenAI Gym results** Table 1 demonstrates that in terms of performance at 500k steps, our LFIW method is able to outperform baseline methods on most tasks (except Hopper-v2 with TD3). On the other hand, PER and ERE do not perform very favorably against uniform sampling; a similar phenomenon has been observed by (Novati & Koumoutsakos, 2018) for PER on other actor-critic algorithms, such as DDPG (Lillicrap et al., 2015) and PPO (Schulman et al., 2017). We also considered combining PER with LFIW, but achieved little initial success. We believe this is the case because PER is designed for $Q$-learning instead of actor-critic methods, where learning the max $Q$-function is the objective.

**DCS results** Table 2 and 4 shows the results with SAC and TD3 on state representations and DrQ on pixel representations. Again, we observe improvements over the baselines in most cases, and comparable performance in others. Notably, we achieve much higher performance with LFIW at 100k training steps, which demonstrates that biasing the replay buffer towards on-policy experiences is able to achieve good policy performance more quickly.

## 6.2 ADDITIONAL ANALYSES

To illustrate the advantage of our method, we perform further analyses over the classification accuracy of $w_\psi$ and the quality of the $Q$-function estimates over the Humanoid-v2 environment trained with SAC and SAC + LFIW. In Appendix C, we also include additional ablation studies over the hyperparameters introduced by LFIW, including temperature $T$, replay buffer size $|\mathcal{D}_f|$ and number of hidden units in $w_\psi$. We observe that SAC + LFIW is insensitive to these hyperparameter changes, which suggests that the same LFIW hyperparameters can be applied readily to other tasks.

---

[2]We use $\alpha = 0.6, \beta = 0.4$ in PER.

**Accuracy of** $w_\psi$    We use $w_\psi$ to discriminate two types of experiences; experiences sampled from the policy trained with SAC for 5M steps are labeled positive, and the mixture of experiences sampled from policies trained for 1M to 4M steps are labeled negative. With the $w_\psi$ predictions, we obtain a precision of $87.3\%$ and an accuracy of $73.1\%$. This suggests that the importance weights tends to be higher for on-policy data as desired, and the weights indeed allows the replay buffer to be closer to on-policy experiences.

**Quality of** $Q$**-estimates**    We compare the quality of the $Q$-estimates between SAC and SAC+LFIW, where we sample 20 trajectories from each policy, and obtain the "ground truth" via Monte Carlo estimates of the true $Q$-value. We then evaluate the learned $Q$-function estimates and compare their correlations with the ground truth values. For the SAC case, the Pearson and Spearman correlations are $0.41$ and $0.11$ respectively, whereas for SAC+LFIW method they are $0.74$ and $0.42$ (higher is better). This shows how our $Q$-function estimates are much more reflective of the "true" values, which explains the improvements in sample complexity and the performance of the learned policy.

## 7    CONCLUSION

In this paper, we propose a principled approach to prioritized experience replay for TD-learning of $Q$-value functions in actor-critic methods, where we re-weigh the replay buffer to be closer to on-policy experiences. We justify this by showing theoretically that when we measure the discrepancy between function with the expected squared differences under state-action distributions, only the on-policy distribution ensures that the Bellman evaluation operator is a contraction for the resulting discrepancy. To achieve a good bias-variance trade-off in practice, we assign weights to the replay buffer based on their estimated density ratios against the stationary distribution. These density ratios are estimated via samples from fast and slow replay buffers, which reflect on-policy and off-policy experiences respectively. Our methods can be readily applied to deep reinforcement learning methods based on actor-critic approaches. Empirical results on SAC, TD3 as well as DrQ on 11 environments and 45 method-task combinations demonstrate that our method based on likelihood-free importance weighting is able to achieve superior sample complexity on most cases compared to other methods, so our approach can be applied as a plug-and-play method to improve actor-critic methods.

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

# A  ALGORITHM

---

**Algorithm 1** Actor Critic with Likelihood-free Importance Weighted Experience Replay

---

1: **repeat**
2:   **for** each environment step **do**
3:     gather new transition tuples $(s, a, r, s')$
4:     update $(s, a, s, s')$ to $\mathcal{D}_s$ (slow replay buffer) and $\mathcal{D}_f$ (fast replay buffer)
5:   **end for**
6:   remove stale experiences in $\mathcal{D}_s, \mathcal{D}_f$ ($|\mathcal{D}_f| < |\mathcal{D}_s|$)
7:   **if** $|\mathcal{D}_s|$ exceeds some threshold **then**
8:     obtain samples from $\mathcal{D}_s$ and $\mathcal{D}_f$
9:     update $w_\psi$ with loss function $L_w(\psi)$ (Eq. 9)
10:     assign $\tilde{w}_\psi$ according to Eq. 10
11:   **else**
12:     $\tilde{w}_\psi = 1$ (no re-weighting)
13:   **end if**
14:   obtain estimates for $B^\pi Q_\theta$ with base algorithm
15:   update $Q_\theta$ with loss function $L_Q(\theta; \mathcal{D}_s, \tilde{w})$ (Eq. 11)
16:   update $\pi_\phi$ and value network (if available) with base algorithm
17: **until** Stopping criterion
18: **return** $Q_\theta, \pi_\phi$

---

# B  PROOFS

**Lemma 1.** *For all $\gamma \in (0, 1)$, the Bellman operator $\mathcal{B}^\pi$ is a $\gamma$-contraction with respect to the $\|\cdot\|_d$ norm if $d = d^\pi$ holds almost everywhere, i.e.,*

$$d = d^\pi \ a.e. \implies \|\mathcal{B}^\pi Q - \mathcal{B}^\pi Q'\|_d \leq \gamma \|Q - Q'\|_d, \forall Q, Q' \in \mathcal{Q}$$

*Proof.* From the definitions of $\|\cdot\|_d$ and $\mathcal{B}^\pi$, we have:

$$\|\mathcal{B}^\pi Q - \mathcal{B}^\pi Q'\|_d^2 \tag{12}$$
$$= \mathbb{E}_{(s,a) \sim d}[(\gamma \mathbb{E}_{s',a'}[Q(s', a')] - \gamma \mathbb{E}_{s',a'}[Q'(s', a')])^2]$$
$$= \gamma^2 \mathbb{E}_{(s,a) \sim d}[(\mathbb{E}_{s',a'}[Q(s', a') - Q'(s', a')])^2]$$
$$\leq \gamma^2 \mathbb{E}_{(s,a) \sim d}[\mathbb{E}_{s',a'}[(Q(s', a') - Q'(s', a'))^2]] \tag{13}$$
$$= \gamma^2 \mathbb{E}_{(s,a) \sim d'}[(Q(s, a) - Q'(s, a))^2] \tag{14}$$
$$= \gamma^2 \|Q - Q'\|_{d'}^2 \tag{15}$$

where $s' \sim P(\cdot|s, a), a' \sim \pi(\cdot|s')$ and

$$d'(s', a') = \sum_{s,a} P(s'|s, a)\pi(a'|s')d(s, a)$$

represents the state-action distribution of the next step when the current distribution is $d$. We use Jensen's inequality over the convex function $(\cdot)^2$ in Eq. 13. Since $d^\pi$ is the stationary distribution, $d = d' \iff d = d^\pi, a.e.$, so the if direction holds.  $\square$

**Theorem 1.** *For all $\gamma \in (0, 1)$, the Bellman operator $\mathcal{B}^\pi$ is a $\gamma$-contraction with respect to the $\|\cdot\|_d$ norm if and only if $d = d^\pi$ holds almost everywhere, i.e.,*

$$d = d^\pi, \ a.e. \iff \|\mathcal{B}^\pi Q - \mathcal{B}^\pi Q'\|_d \leq \gamma \|Q - Q'\|_d, \forall Q, Q' \in \mathcal{Q}$$

*Proof.* The "if" case is apparent from the Lemma, so we only need to prove the "only if" case.

For the "only if" case, we construct a counter-example for all $d \neq d'$. Without loss of generality, assume $\forall s, a \in \mathcal{S} \times \mathcal{A}, Q'(\boldsymbol{x}) = 0$. The following functionals of $Q$

$$h(Q) := \|\mathcal{B}^\pi Q - \mathcal{B}^\pi Q'\|_d^2 / \gamma^2 = \mathbb{E}_{(s,a) \sim d}[(\mathbb{E}_{s',a'}[Q(s', a')])^2]$$

$$g(Q) := \|Q - Q'\|_d^2 = \mathbb{E}_{(s,a) \sim d}[Q(s,a)^2]$$

corresponds to the quantities at the two ends of the contraction argument. Our goal is to find some $Q \in \mathcal{Q}$ such that $h(Q) - g(Q) > 0$, which would complete the contradiction. We can evaluate the functional derivatives for $h(Q)$ and $g(Q)$:

$$\frac{\mathrm{d}h}{\mathrm{d}Q}(s', a') = 2 \sum_{s,a} d(s,a) \mathcal{E}(s,a) P(s'|s,a) \pi(a'|s')$$

$$\frac{\mathrm{d}g}{\mathrm{d}Q}(s, a) = 2d(s,a)Q(s,a)$$

where $\mathcal{E}(s,a) = \mathbb{E}_{s'' \sim P(\cdot|s,a), a'' \sim \pi(\cdot|s'')}[Q(s'', a'')]$ is the expected $Q$ function of the next step when the current step is at $(s, a)$. Now let us consider some $Q_0$ such that for some constant $q > 0$, $\forall s, a \in \mathcal{S} \times \mathcal{A}, Q_0(s,a) = q$. Let us then evaluate both functional derivatives at $Q_0$:

$$\left.\frac{\mathrm{d}h}{\mathrm{d}Q}(s', a')\right|_{Q_0} = 2 \sum_{s,a} d(s,a) q P(s'|s,a) \pi(a'|s') = 2d'(s', a')q$$

$$\left.\frac{\mathrm{d}g}{\mathrm{d}Q}(s, a)\right|_{Q_0} = 2d(s,a)q$$

where $\mathcal{E}(s,a) = q$ under the current $Q_0$. Because $d'$ and $d$ are not equal almost everywhere (from the assumption of $d$ not being the stationary distribution), there must exist some non-empty open set $\Gamma \in \mathcal{S} \times \mathcal{A}$ where $\int_\Gamma (d'(s,a) - d(s,a)) \, \mathrm{d}s \, \mathrm{d}a > 0$. We can then add a function $\epsilon : \mathcal{S} \times \mathcal{A} \to \mathbb{R}$ such that $\epsilon(s,a) = \nu \cdot \mathbb{I}((s,a) \in \Gamma)$ where $\mathbb{I}$ is the indicator function and $\nu$ is an infinitesimal amount. Now let us evaluate $(h - g)$ at $(Q_0 + \epsilon)$:

$$(h - g)(Q_0 + \epsilon)$$

$$= (h - g)Q_0 + \left(\frac{\mathrm{d}h}{\mathrm{d}Q} - \frac{\mathrm{d}g}{\mathrm{d}Q}\right)\epsilon + o(\nu)$$

$$= (q^2 - q^2) + 2q \int_\Gamma (d'(s,a) - d(s,a))\nu \, \mathrm{d}s \, \mathrm{d}a + o(\nu) > 0$$

Therefore, the proposed function $(Q + \epsilon)$ is the contradiction we need. $\qquad\square$

## C ADDITIONAL EXPERIMENTAL DETAILS

### C.1 SETUP ON CHAIN MDP

The chain MDP considered has deterministic transitions, so we make the policy stochastic to make sure that a stationary distribution exists. In each epoch over all the state-action pairs, we use the following TD learning update over tabular data to simulate the effect of weighting with fixed learning rate $\eta$:

$$Q(s,a) \to Q(s,a) + (1 - (1 - \eta)^{w(s,a)})(\mathcal{B}^\pi Q(s,a) - Q(s,a)) \tag{16}$$

where $w(s,a)$ is the weight (uniform, TD error or $d^\pi$) which simulates the number of TD updates with learning rate $\eta$. The weights are normalized to have a mean value of 1 which makes number of updates per epoch the same across different methods.

### C.2 ABLATION STUDIES

To demonstrate the stability of our method across different hyperparameters, we conduct further analyses over the key hyperparameters, including the temperature $T$ in Eq. 10, the size of the fast replay buffer $|\mathcal{D}_\mathrm{f}|$, and the number of hidden units in the classifier model $w_\psi$. We consider running the SAC+LFIW method on the Walker-v2 environment with 1000 episodes using all the default hyperparameters unless explicitly changed.

Table 5: Additional hyperparameters for SAC (Haarnoja et al., 2018)

| Parameter | Value |
|---|---|
| optimizer | Adam Kingma & Ba (2014) |
| learning rate | $3 \times 10^{-4}$ |
| discount | 0.99 |
| number of samples per minibatch | 256 |
| nonlinearity | ReLU |
| target smoothing coefficient | $5 \times 10^{-3}$ |

Table 6: Additional hyperparameters for TD3 (Fujimoto et al., 2018)

| Parameter | Value |
|---|---|
| optimizer | Adam Kingma & Ba (2014) |
| learning rate | $10^{-3}$ |
| discount | 0.99 |
| number of samples per minibatch | 256 |
| nonlinearity | ReLU |
| exploration policy | $\mathcal{N}(0, 1)$ |

**Temperature $T$**  The temperature $T$ affects the variances of the weights assigned; a larger $T$ makes the weights more similar to each other, while a smaller $T$ relies more on the outputs of the classifier. Since we are using finite replay buffers, using a larger temperature reduces the chances of negatively impacting performance due to $w_\psi$ overfitting the data. We consider $T = 1, 2.5, 5, 7.5, 10$ in Figure 2a; all cases have similar sample efficiencies except for $T = 1$. Similarly, we also perform a similar analysis on Humanoid-v2 with SAC in Figure 3. We observe a similar dependency on $T$ as in Walker where the sample efficiency with $T = 1$ is significantly worse that for the other hyperparameters considered, which shows that overfitting the data can easily be avoided by using a higher temperature value even in higher-dimensional state-action distributions.

**Replay buffer sizes $|\mathcal{D}_f|$**  The replay buffer sizes $|\mathcal{D}_f|$ affects the amount of experiences we treat as "on-policy". Larger $|\mathcal{D}_f|$ reduces the risk of overfitting while increasing the chances of including more off-policy data. We consider $|\mathcal{D}_f| = 1000, 10000, 50000, 100000$, corresponding to 1 to 100 episodes. We note that $|\mathcal{D}_s| = 10^6$, so even for the largest $\mathcal{D}_f$, $\mathcal{D}_s$ is significantly larger. The performance are relatively stable despite a small drop for $|\mathcal{D}_f| = 100000$.

**Hidden units of $w_\psi$**  The number of hidden units at each layer affects the expressiveness of the neural network. While networks with more hidden units are more expressive, they are easier to overfit to the replay buffers. We consider hidden layers with $128, 256$ and $512$ neurons respectively. While the smaller network with $128$ units is able to achieve superior performance initially, the other configurations are able to catch up at around 1000 episodes.

Table 7: Additional hyperparameters for DrQ (Kostrikov et al., 2020)

| Parameter | Value |
|---|---|
| optimizer | Adam Kingma & Ba (2014) |
| learning rate | $10^{-3}$ |
| discount | 0.99 |
| critic target update frequency | 2 |
| actor log stddev bounds | [-10, 2] |
| actor update frequency | 2 |
| number of samples per minibatch | 512 |
| nonlinearity | ReLU |
| image dimensions | $84 \times 84$ |

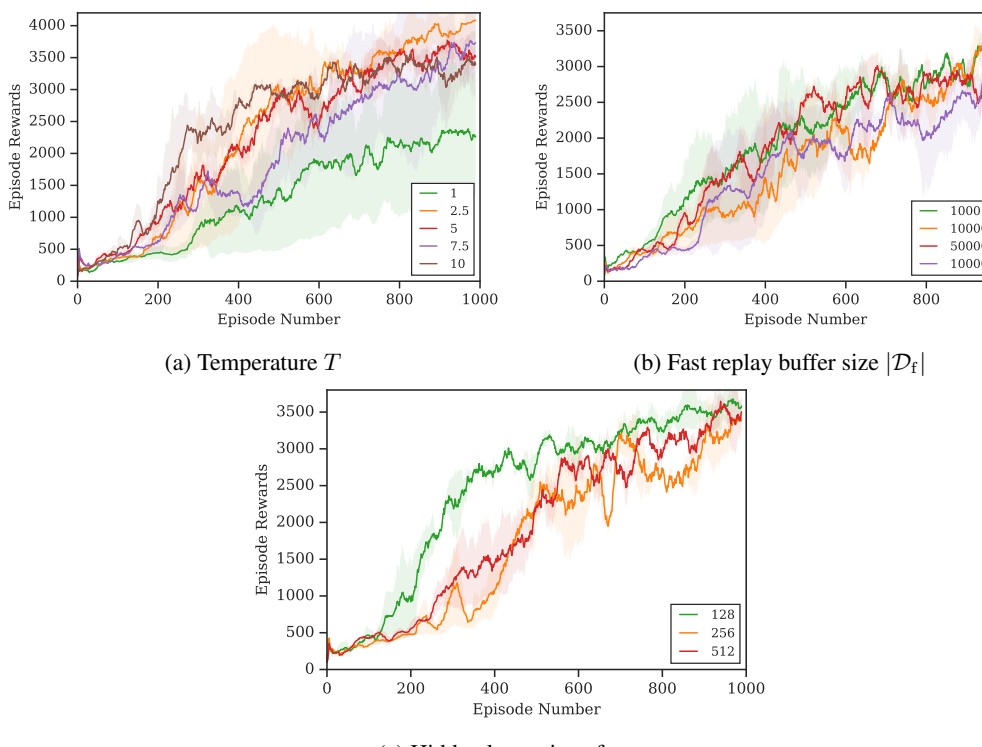

(a) Temperature $T$    (b) Fast replay buffer size $|\mathcal{D}_\mathrm{f}|$

(c) Hidden layer size of $w_\psi$

Figure 2: Hyperparameter sensitivity analyses on Walker2d-v2 with SAC.

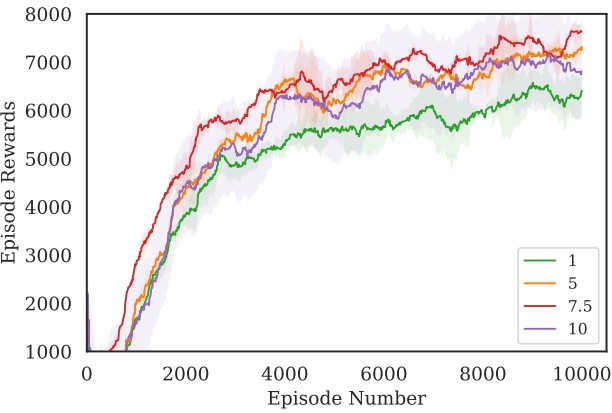

Figure 3: Temperature sensitivity on Humanoid-v2 with SAC

