# OpenReview forum: "Experience Replay with Likelihood-free Importance Weights"
_ICLR.cc/2021/Conference — Reject_

### Official Review · AnonReviewer1 · 2020-10-28
**FA or tabular, continuous or discrete?**

**Rating:** 3
**Confidence:** 4

**Review:**

The paper proposed a method to address the distribution mismatch problem in off-policy learning. Its method is to estimate the density ratio between the stationary distribution under the target policy and the data distribution in the replay buffer, and then correct the state distribution with the ratio so that it can use experience from the replay as if they were generated from the target policy.

Although I have many questions about the paper, the following two are most important currently and are hoped to be addressed first.

It seems not clear to me whether this paper considers the tabular case or the function approximation case. It wrote in section 5 that "our goal is to improve performance of TD learning with function approximation", but its preliminaries (Section 2), its example (Figure 1), and its main theoretical result (Theorem 1) are all for the tabular case. In fact, the state distribution mismatch only matters when multiple states share the same parameter. For the tabular case, as each state has its own parameter, there is no interference between them, and thus there is no need to make the state distribution on-policy.

Empirically, all the experiments are designed to solve continuous control problems, where both state and action are continuous. However, the density ratio d^\pi(s, a) / d^D(s, a) is only defined under the discrete setting. In fact, because both state and action spaces are continuous, all elements in the replay buffers are different from others and this ratio makes no sense. Therefore this algorithm doesn't seem appropriate to me for continuous control tasks.

---

> ### Author Response · Authors · 2020-11-20
> **Response: density ratios in continuous control tasks**
>
> Thank you for your time to review the paper and for constructive comments. We address the points raised regarding density ratio below.
>
>
> Q: “In fact, the state distribution mismatch only matters when multiple states share the same parameter.”?
>
> A: We note that mismatch matters even in the tabular case. Since we are interested in how to estimate the on-policy expectation from off-policy experiences via important sampling, we are discussing state distribution mismatch between off-policy and on-policy experiences. As long as on-policy is not equal to off-policy, the state-action distributions do not match, and this matters because correctly estimating the ratio will reduce bias; it is not only an issue of function approximation.
>
> Q: Density ratio d^\pi(s, a) / d^D(s, a) is only defined under the discrete setting, why is it valid under the continuous case?
>
> A: In the discrete case, this is the ratio of probability mass functions, and in the continuous case, this is the ratio of probability density functions. For example, if $d^\pi$ is Gaussian $\mathcal{N}(0, 1)$ and $d^D$ is Gaussian $\mathcal{N}(1, 1)$, then $d^\pi / d^D$ at point $x$ exists and it equals $exp(-0.5 * (2x - 1)) / \sqrt{2\pi}$.
>
> We refer to two (amongst many) influential papers that estimates from samples the ratio between two probability density functions (pdf) over continuous spaces:
>
> [Goodfellow et al. 2014] Generative Adversarial Networks. Proposition 1 clearly states that the optimal discriminator is a density ratio between the data distribution and the mixture of data and generated distribution, which are typically over continuous sample space.
>
> [Ho and Ermon., 2016] Generative Adversarial Imitation Learning. Corollary A 1.1. They clearly used such an approach for many continuous control tasks.

---

> ### Author Response · Authors · 2020-11-23
> **Discussion**
>
> Kindly let us know if our response below addressed your concerns. We are happy to answer if there are additional issues/questions.

---

### Official Review · AnonReviewer4 · 2020-10-29
**Mixed Impressions, early problems with related work but overall appealing approach**

**Rating:** 7
**Confidence:** 4

**Review:**

The authors propose likelihood-free importance weights applied to
experience replay buffers in reinforcement learning. When the sampling
distribution is not the on-policy distribution the RL agent may learn
a biased estimate. This could happen in a number of ways, but two
obvious ones are data being generated off-policy or prioritized
experience replay (PER). In both cases the sampling distribution is
not the one we wish to minimize over. In PER this is partly fixed by
the importance sampling correction. This work replaces this
correction, in the off-policy setting, with a learned ratio.

Although exposition and some of the early details about related work
and background seem problematic, the experimental results are strong
and encouraging. The fundamental idea is broadly applicable, although
some of the related work touches on related ideas this particular
application of the idea appears novel and potentially impactful.

-- Pros:

Broadly useful technique for LFIW, and specific application using the
two replay buffers is quite nice.
Relatively strong empirical results

-- Cons:

Issues with the background / related work (see comments below)
Experiments could be presented and discussed more clearly

-- Comments:
"However, such a heuristic could be highly sub-optimal in actor-critic
methods..."
This sentence and the rest of the paragraph seem to be missing
something essential. Prioritized experience replay is an importance
sampling technique. That is, while sampling under a biased
distribution, they use importance weights to correct for this sampling
bias (only partly in the case of PER in practice). As well, it is not
just actor-critic methods that require the expected loss to be under a
particular distribution over states.

This misunderstanding reappears just before section 3, where the
authors discuss PER as a heuristic for improved performance by
changing the expectation in Equation 2. This is, indeed, incorrect. In
terms of Equation 2, we could accurately represent PER by saying that
D is already reweighted by the priorities (which is how the authors
currently interpret it), and further adding a reweighting where inside
the expectation we divide by something like p_D(s, a), to convert the
expectation into one over the uniform distribution. The most accurate
way obviously would be the seperate these two out explicity: a
sampling distribution (data generated by a behavior policy and put
into replay) and an importance sampling distribution (the prioritized
distribution), and multiplying inside the expectation by their ratio.

It is strange that these errors would crop up considering the primary
contribution of the work is about essentially doing this without
explicitly computing this ratio of likelihoods. The lack of correction
re-appears in equation 3 as well

Finally, this comes full circle just before section 3.1 with the
statement that the 'prioritized' distribution needs to be d^\pi. Of
course it has to be d^\pi when you do not correctly correct for the
sampling distribution.
Overall, I don't want to attribute this to an actual
'misunderstanding' by the authors, I think section 4 clearly shows the
depth of their understanding. So instead, I think of this as an issue
with the writing and how things are explained. However, how PER is
communicated here does make me fairly wary about the empirical results
for PER.

Regarding Theorem 1, one direction of this has been known for quite a
long while. That is, the contraction holds when the distribution is
d^\pi. So, it might be better to spend more time emphasizing the other
direction and discussing its proof.

"In actor-critic methods, the critic objective typically estimates the
Q-value function"
Arguably the most typical estimate for actor-critic would be the value
function (V) not the action-value function (Q). Not that AC methods
don't ever use Q-values, but this doesn't seem like the typical case.


-- Minor suggestions / Typos:

Page 2, just above equation 1: "based on Bellman equation Q", missing a "the"

Footnote 1 on page 3 should probably come earlier, during
preliminaries as it is not specific to PER.

Page 5, last paragraph, "We observe that using the weights to correct
the policy updates does not demonstrate provide much..." (remove
either 'demonstrate' or 'provide')

I struggled to find where the authors specify the f-divergence they use.

-- Questions:

Did you consider using your technique combined with prioritized
replay? This would mean changing the 'slow' replay buffer
probabilities to be prioritized probabilities, but otherwise would be
identical to what is currently shown in the paper. More specifically,
although the authors compare against PER as though this is an
alternative to PER, it seems more fitting to think of this as an
alternative to the IS correction being used by PER (among many other
methods).

The authors report the accuracy of w_\psi used as a discriminator,
which is great because I was going to ask about this exactly before I
got to that part of the results. However, I wonder about a few
additional details here. Have you looked at how this
precision/accuracy varies over training? In particularly, you might
expect that it becomes nearly impossible to discriminate as the policy
convergences and thus the two distributions slowly become identical.

Also, how robust is performance to degradations in this accuracy? This
seems like something you might have looked at. If so, do you have any
insights? The reported accuracy is low enough, with empirical
performance strong enough, that I suspect it *is* quite robust.

---

> ### Author Response · Authors · 2020-11-20
> **Response: New experiments with PER and LFIW and discussions on PER**
>
> Thank you for your time to review the paper and for constructive comments. We address each points individually.
>
>
> Q: About PER and importance sampling: is the proposed method simply a different way of implementing PER?
>
> A: We absolutely agree with your argument with regards to PER being an important sampling algorithm, and we have made the edits in the revisions to emphasize the point.
> That being said, we believe that our method is critically different from PER in terms of the target expectation that we are trying to estimate with samples; our argument is as follows.
> Suppose we have some off-policy distribution $P$, and our goal is to estimate the expectation of a function $h$. Let $Q$ denote the on-policy distribution for the current policy.
> In LFIW, our target is the “on-policy distribution”, which is $E_Q[h]$,  and we do importance sampling by $\mathbb{E}_P[w \cdot h]$, where the weights $w$ is simply $Q / P$.
> In PER, however, we have the prioritization distribution $R$, which is $P$ times the sampling prioritization weights $s$ ($P(i)$ in PER paper), as well as the weights $w$ that are inversely correlated with $s$ (assuming that $\beta = 1$ in the PER paper). So the importance sampling is $\mathbb{E}_R[w \cdot h] = \mathbb{E}_P[s \cdot w h] = \mathbb{E}_P[h]$. The last step assumes that $\beta = 1$, so $w = 1/s$, and this corresponds to uniform sampling from the offline replay buffer $P$.
> This means that PER is estimating the expectation under $P$ (off-policy, at least for $\beta = 1$, for other $\beta$ values it is unclear what expectation PER is exactly trying to estimate), and ours is estimating the expectation under $Q$ (on-policy). And the benefit of PER in Q-learning is variance reduction, whereas our focus here is bias reduction.
> Therefore, PER works well for max-Q functions, but not necessarily for Q-value functions of the current policy. This is not only observed by our work, but also another method [Wang and Ross, 2019] (this is also a baseline in the manuscript) , which said that “We found that it can be relatively difficult to find a good hyperparameter combination for SAC+PER.”
>
> [Wang and Ross, 2019] Boosting soft Actor-Critic: Emphasizing Recent Experience Without Forgetting the Past
>
>
> Q: How does the method perform when combined with PER?
>
> A: As we explained earlier, our method is fundamentally different from PER in terms of the expectation being evaluated (off-policy in PER, verses nearly on-policy with LFIW). Nevertheless, we perform another experiment that estimates the on-policy expectation with PER, which we show in Table 3 of the revision, where we see that the performance of just adding LFIW remains the best, but adding LFIW and PER together, improves the performance of the agent above the vanilla agent and just adding PER.
> Concretely, we resample the slow replay buffer with the sampling procedure of PER, and estimate the likelihood ratio with LFIW. This still estimates the expectation of $E_Q[h]$, since the likelihood ratios are obtained for $Q/R$ instead of $Q/P$. In Table 3 of the revision, we show that PER+LFIW outperforms PER but performs similarly to LFIW; this shows that the improvement comes from correcting for the bias in the objective, and not necessarily the variance.
>
>
> Q: Theorem 1’s “if” part is not that significant?
>
> A: In the revision, we have separated Theorem 1 into Lemma 1 and Theorem 1, which discuss the “if” and “only if” directions respectively.
>
>
> Q: What are the used f-divergences?
>
> A: The one we use is the Jensen-Shannon divergence. We mentioned this right before Section 6.1 in the original version, and we add an additional note about this in Section 4 of the revision.

---

> > ### Comment · AnonReviewer4 · 2020-11-24
> > **Thank you for your response**
> >
> > Thank you for clarifying in your first two responses, this really helped me see where I was misunderstanding the work. I find the results in Table 3, combining PER and LFIW, interesting. There are several possible interpretations for the lack of benefit of adding PER, such as PER improving performance for a reason different from what is usually purported or the use of prioritization making the problem faced by LFIW harder in some way. Any insight you might offer here would be beneficial (to me and to other readers I suspect).

---

> > > ### Author Response · Authors · 2020-11-24
> > > **More discussions on PER and LFIW**
> > >
> > > A: Thank you for the response!
> > >
> > > We believe that the major benefit of PER comes from variance reduction, since it prioritizes resampling and the downweights the samples that are more frequently sampled. This strategy would work if we are trying to learn the optimal Q function (where the distribution of state-action pairs of the current policy does not really matter). However, the quantity of interest in our paper is the Q value function under the current policy, so sampling / reweighting more on-policy experiences more frequently seems more helpful as it benefits us thorough bias reduction (especially when we have a limited amount of compute to perform the updates to the Q value function approximator). Combining PER+LFIW gives us best of both worlds (we have a less biased objective with potentially less variance); based on our arguments earlier, intuitively we would expect that bias(PER) > bias(LFIW) \approx bias(PER+LFIW), and var(LFIW) > var(PER+LFIW) \approx var(PER). This also explains why PER works very well on Q-learning, but not nearly as well on actor-critic methods.
> > >
> > > Our experiments show that PER+LFIW outperforms PER but is on par with LFIW, which means that the benefit in the actor-critic case comes mostly from bias reduction (even though variance reduction does not hurt performance). We will add a more detailed discussion in the next version.

---

> ### Author Response · Authors · 2020-11-23
> **Discussion**
>
> Kindly let us know if our response below addressed your concerns. We are happy to answer if there are additional issues/questions.

---

### Official Review · AnonReviewer2 · 2020-10-29
**Learnable re-weighting of samples for actor-critic algorithms**

**Rating:** 5
**Confidence:** 4

**Review:**

The paper proposes a generally applicable modification to experience sampling in the context of actor-critic algorithms using a Q function as a critic. The modification is called "Likelihood-free Importance Weights" (LFIW). The authors describe the approach in Appendix A in the form of pseudocode.  Comparing to a generic actor-critic algorithm, the changes include the keeping of two replay buffers ("fast" and "slow") and inclusion of an additional re-weighting function w which in turn is used in the update of the Q function. The paper includes a thorough performance comparison on MuJoCo and DM Control Suite.

The results are good, but the authors seem to use a weak implementation of SAC. For comparison, I am referring to SAC as implemented in https://github.com/tensorflow/agents/blob/v0.6.0/tf_agents/agents/sac/sac_agent.py#L62-L634  Using this implementation I am getting, e.g. for the Humanoid-v2 environment,  at 500K steps results above 4000, compared to 3189 (SAC+LFIW) and 2033 (SAC) reported in the paper. Hence it is hard for me to assess whether LFIW offers a real improvement of the SOTA or perhaps fixes some problems of the underlying implementation of SAC.

The mathematical analysis contained in Theorem 1 is interesting, but in my opinion, it is written confusingly. It would be better to decompose it into two separate statements:
- the first statement stating the inequality with a simple proof based on the convexity of the square function,
- the second statement proposing a counter-example in the form of Q+epsilon for appropriately small epsilon.

Also, the statement of the theorem is slightly weaker than I would like: can we just prove, that the counterexample exists regardless of gamma? The current statement says that the mapping is not a gamma-contraction, but one can imagine, that relaxing gamma would still lead to a contractive mapping.

The example presented in Figure 1 is interesting, though in my opinion, somewhat detached from the main focus of the paper which in my opinion is the analysis of Algorithm 1. Also, the three-state MDP may seem too simple to conclude about the performance of the sampling method.

---

> ### Author Response · Authors · 2020-11-20
> **Author response: clarifications on implementation, theorem 1 and toy MDP setting.**
>
> Thank you for your time to review the paper and for constructive comments. We address each points individually.
>
> Q: Weaker implementation of SAC for Humanoid
>
> A: In the SAC paper, the results reported for Humanoid use a different set of hyperparameters than each of the other environments. In our SAC implementation, we use the same set of hyperparameters for all the environments for the SAC agent.  It is likely to improve the baseline Humanoid-v2 SAC agent with and without LFIW using the new set of hyperparameters. The set of hyperparameters used in this paper are the ones that were utilized for all the environments except humanoid, in the original SAC paper. On the other hand, our method also improves over other methods, such as the recent DrQ that has achieved high performance for control in image space.
>
> Q: Regarding the statements in Theorem 1
>
> A: Thank you for the suggestion. In the revision, we separated the “if” and “only-if” statements into Lemma 1 and Theorem 1 respectively, and updated the arguments about $\gamma$. We believe that our argument holds for all $\gamma$ between 0 and 1.
>
> Q: Also, the three-state MDP may seem too simple to conclude about the performance of the sampling method.
>
> A: The toy MDP case only serves to show that the idea of reweighting makes sense in a tabular scenario, where we can estimate the likelihood ratios without issue. Our empirical focus remains on the OpenAI Gym and DeepMind Control Suite tasks, where we have demonstrated that the same idea works also for high-dimensional control tasks (such as DrQ on DeepMind Control Suite).

---

> ### Author Response · Authors · 2020-11-23
> **Discussion**
>
> Kindly let us know if our response below addressed your concerns. We are happy to answer if there are additional issues/questions.

---

### Official Review · AnonReviewer3 · 2020-10-30
**Good paper with a few points not clearly described**

**Rating:** 6
**Confidence:** 4

**Review:**

This paper is on an experience replay approach, as applied to deep RL methods, that uses a density ratio between on-policy and off-policy experiences as the prioritization weights. The objective is to find appropriate bias-variance trade-offs for importance sampling from the replay buffer. In particular, there's the bias issue from replay experiences of other policies, and the variance issue from the recent on-policy experiences.

It's not entirely clear to the reviewer about the necessity of maintaining two replay buffers (slow and fast). Instead of maintaining two replay buffers, it seems one can simply retain the standard single-buffer strategy, and evaluate how likely the experience is with respect to the current policy. Then the likelihood can be used as the weight for prioritized experience replay. This simple strategy also takes the bias-variance trade-offs coming from on-policy and off-policy experiences. The reviewer is curious about the advantage of the developed approach that uses two buffers over what's described above.

In the paper, the slow buffer is considered for maintaining off-policy experience, and the fast buffer is for on-policy experience. Accordingly, the sizes of those two buffers are supposed to be very important parameters. For instance, if the two buffers have similar sizes, then the developed approach are expected to function like standard deep RL (on policy or off policy depending on the buffer size). However, Figure 2 (b) shows that the performance is not sensitive to the size of the fast (on-policy) buffer. The result is counter-intuitive, though it's clear that the results are supposed to show the insensitivity to such parameters.

Experiments were conducted by combining the experience replay approach (called LFIW, likelihood-free importance weighting) with three existing deep actor-critic methods, and then comparing the combinations with their originals. The results look good, and demonstrate the effectiveness of the developed approach. It's unclear why the results were presented in tables instead of curves (minor point), which can be potentially more better for readability.

---

> ### Author Response · Authors · 2020-11-20
> **Author response: implementation explanations**
>
> Thank you for your time to review the paper and for constructive comments.
>
>
> Q: Can we use a single buffer, where we use likelihoods with respect to the current policy?
>
> A: Implementation-wise, we can use a single buffer, where the fast and slow only comes from the size of the buffer (e.g. the fast buffer tracks the previous 10k state-action pairs, whereas a slow buffer tracks the previous 1M state-action pairs). It is possible to evaluate $\pi(a | s)$ when the policy is stochastic, but the same cannot be said for deterministic policies (such as TD3). Evaluating the likelihood of $(s, a)$ under the stationary distribution would be much more challenging even for stochastic policies, as one has to marginalize over all possible paths that reach that state-action pair.
>
>
> Q: How does the performance not depend on the size of the fast buffer?
>
> A: The fast buffer we considered in Figure 2(b) is much smaller than the slow one. The biggest buffer contains 100 episodes and is 10% of the size of the slow buffer. If the fast buffer has almost the same size as the slow one, then it would indeed be difficult to achieve superior performance than the baseline, as it amounts to no reweighting at all.
>
>
> Q: Why not present the results in curves?
>
> A: We considered the option, and decided that tables are more concise, since we have 45 environment-task combinations, and each will correspond to a different method. Moreover, sometimes the reward function may not be entirely well-specified, which makes it challenging to highlight the relative gain of the method.

---

> ### Author Response · Authors · 2020-11-23
> **Discussion**
>
> Kindly let us know if our response below addressed your concerns. We are happy to answer if there are additional issues/questions.

---

### Author Response · Authors · 2020-11-20
**Author response**

We thank each of the authors for their time to review the paper. We have replied to each others concerns individually to clarify points regarding Prioritized Experience Replay (PER), density ratios in continuous control tasks, and some implementation details. We have also included new experiments with LFIW and PER together in Table 3, as per Reviewer 4's suggestion.

---

### Decision · Program_Chairs · 2021-01-07
**Final Decision**

**Decision:**

Reject

**Comment:**

This paper proposed a new experience replay approach, applicable to deep RL methods. Two reviewers suggested acceptance and two did rejection. The first negative reviewer R1 raised a concern on continuous vs. discrete issue, but AC thinks that the authors' response is not fully convincing enough.  The second negative reviewer R2 pointed out that the reported performance of SAC is poor compared to the existing implementation (although authors claim a different set of hyperparameters is used), which AC thinks is a critical weakness to judge the value of the experiments. Two other positive reviewers (even R4) shows mixed opinions. Overall, AC thinks this is a borderline paper, a bit toward rejection.